# Clinical and Neurophysiological Effects of Botulinum Neurotoxin Type A in Chronic Migraine

**DOI:** 10.3390/toxins13060392

**Published:** 2021-05-29

**Authors:** Mariarosaria Valente, Christian Lettieri, Valentina Russo, Francesco Janes, Gian Luigi Gigli

**Affiliations:** 1Clinical Neurology Unit, Santa Maria della Misericordia University Hospital, 33100 Udine, Italy; mariarosaria.valente@uniud.it (M.V.); christian.lettieri@asufc.sanita.fvg.it (C.L.); valentina.russo@asufc.sanita.fvg.it (V.R.); gianluigi.gigli@uniud.it (G.L.G.); 2Department of Medicine, University of Udine, 33100 Udine, Italy; 3Intensive Rehabilitation Unit, Santa Maria della Misericordia University Hospital, 33100 Udine, Italy

**Keywords:** chronic pain, transcranial magnetic stimulation, chronic headache, migraine

## Abstract

Chronic pain syndromes present a subversion of both functional and structural nociceptive networks. We used transcranial magnetic stimulation (TMS) to evaluate changes in cortical excitability and plasticity in patients with chronic migraine (CM) treated with botulinum neurotoxin type A (BoNT/A). We enrolled 11 patients with episodic migraine (EM) and 11 affected by CM. Baseline characteristics for both groups were recorded using single- and paired-pulse TMS protocols. The same TMS protocol was repeated in CM patients after four cycles of BoNT/A completed in one year. At baseline, compared with EM patients, patients with CM had a lower threshold in both hemispheres (right hemisphere: 46% ± 7.8 vs. 52% ± 4.28, *p* = 0.03; left hemisphere: 52% ± 4.28 vs. 53.54% ± 6.58, *p* = 0.02). In EM, paired-pulse stimulation elicited a physiologically shaped response, whereas in CM, physiological intracortical inhibition (ICI) between 1 and 3 ms intervals was absent at baseline. On the contrary, increasing intracortical facilitation (ICF) was observed for all interstimulus intervals (ISIs). In CM, cortical excitability was partially reduced after BoNT/A treatment, along with a significant decrease observed in MIDAS score (from 20.7 to 9.8; *p* = 0.008). The lower motor threshold in CM reflects a higher cortical hyperexcitability. The lack of physiological ICI in CM could indicate sensitisation of the trigeminovascular system. Although reduced, this type of response is still observable after treatment, despite a marked clinical improvement. Our study suggests a long-term alteration of cortical plasticity due to chronic pain.

## 1. Introduction

Acute pain lasts for only a limited amount of time and is produced by tissue injury and the concurrent activation of local nociceptive transducers. It is usually related to trauma or invasive procedures, or it appears as a symptom during the course of some pathological processes. Pain is defined as chronic if it lasts for a long period of time and does not respond completely to therapy. Chronic pain may also be initiated by local injury or disease, but it usually persists for a longer period of time and tends to be maintained by factors not directly linked to the original event [1]. The process of transformation from episodic to chronic headaches is not dissimilar to the events that occur when painful conditions affecting other parts of the body become chronic [2,3,4].

Neuroimaging and non-invasive brain stimulation techniques allow the identification of a complex network of brain structures that contribute to the pain experience and their specific roles in each dimension of the whole phenomenon. Most of these brain areas are multimodal, responding to both noxious and salient non-noxious stimuli. The primary relay nodes of this network include the primary and secondary somatosensory cortices, cingulate cortex, posterior parietal cortex, and prefrontal cortex. However, the thalamus, insula, several brainstem structures, and other interconnected brain areas take part in this network as well.

The International Classification of Headache Disorders (ICHD-3) defines CM as a headache that occurs on 15 or more days per month for at least 3 months and has the features of a migraine headache at least 8 days per month [5]. Typically evolving from EM, CM affects 2–3% of the general population and provokes extreme disability. The chronic form tends to be refractory to both symptomatic and prophylactic treatments. The evolution from EM to CM is not completely understood, but several mechanisms seem to be involved, including central sensitisation, reduced descending pain inhibitory control, and cortical hyperexcitability [6]. In particular, a state of sustained cortical brain hyperexcitability has been described between migraine attacks [7] and, more generally, patients with chronic pain present a subversion of both functional and sometimes structural nociceptive networks [8]. Non-invasive brain stimulation techniques based on magnetic fields (transcranial magnetic stimulation, TMS) have been shown to be safe and effective tools to explore cortical excitability, activation, and plasticity in migraine. In particular, single-pulse TMS (sp-TMS) and paired-pulse TMS (pp-TMS) have been used to assess cortical excitability and plasticity, whereas repetitive TMS has been used to modulate cortical plasticity in patients with migraine [9,10].

In 2010, the FDA approved botulinum toxin type A (BoNT/A) for the prophylactic treatment of CM [11]. BoNT/A has proved to be effective in the reduction of headache frequency and severity in patients with CM. The treatment is generally well tolerated by the patient. This therapy is administered through intramuscular injections across seven specific head and neck muscles, repeated at intervals of 12 weeks for 4 cycles [11]. Initially, it was thought that BoNT/A had only a local effect by inhibiting neuromuscular transmission, but recent animal studies suggest that this toxin can act in other areas by both retrograde and anterograde transport in axons [12,13,14,15]. The mechanisms behind the effect of BoNT/A in CM may also include modulation of neurotransmitter release, changes in surface expression of receptors and cytokines, and the enhancement of opioidergic transmission. Some studies demonstrated that BoNT/A acts in migraine by reversibly attenuating neuropeptide and neurotransmitter exocytosis (substance P, calcitonin gene-related peptide, and glutamate) from peripheral sensory neurons, thus obtaining a direct inhibition of peripheral sensitisation; this peripheral inhibition leads to an indirect reduction of central sensitisation [16].

The primary aim of this study was to evaluate cortical excitability in a group of patients affected by CM at baseline and after repeated injections of BoNT/A lasting one year; as a secondary endpoint, we compared the neurophysiological parameters obtained in CM patients (at baseline) with those of patients affected by EM.

## 2. Results

We initially selected 20 patients with CM without aura. Five CM patients were excluded because they were treated with antiepileptic drugs or benzodiazepines or because of contraindications for TMS; four additional CM patients denied their consent to BoNT/A treatment or the TMS protocol. At the end of the selection process, 11 patients with CM without aura were enrolled in the study. Eleven patients affected by EM without aura were then selected among the personnel of the same centre who volunteered to participate.

Nine out of eleven patients (81.8%) were female in both the EM and CM groups. The mean age was 32.4 years in the EM group and 36.5 years in the CM group (*p* = 0.17). Three CM patients dropped out of the study: one patient decided to discontinue BoNT/A treatment and two patients did not accept repeating the study protocol. None of the EM patients were taking drugs for migraine prophylaxis at the time of the study. In the CM group, two patients were treated with SNRI (venlafaxine) at baseline, and the treatment was not interrupted during the follow-up period.

Table 1 and Figure 1 describe the comparison of the neurophysiological variables between EM and CM patients as well as in CM patients between baseline and after BoNT/A treatment. Comparing EM and CM patients, we observed a significantly lower RMT in CM for both cerebral hemispheres, whereas we found no significant differences in CSP or CMT (Table 1A). RMT in CM patients was not modified by BoNT/A treatment (Table 1B). At baseline, pp-TMS exhibited a significant lack of physiological SICI between 1 and 4 ms in CM patients; conversely, in the same interval, a facilitating response occurred. Facilitation was maintained and further amplified for ISIs of 6, 10, and 15 ms. This “paradoxical” facilitation at ISIs of 1–4 ms was partially, although not significantly, reduced by BoNT/A treatment. From a clinical point of view, after BoNT/A treatment, we also observed a marked reduction of pain in CM, with the mean MIDAS score decreasing from 20.7 to 9.8 (*p* = 0.008). A post-hoc power analysis revealed low to moderate power (0.20–0.40) for the pp-TMS variations in CM between pre-treatment and post-treatment measurements as well as a moderate to high power (0.71) for sp-TMS parameters between EM and CM.

## 3. Discussion

TMS is an indirect, non-invasive, and painless method for evaluating cortical excitability and plasticity. The electrical current induced by TMS results in a direct transmembrane depolarisation in the susceptible axons. TMS-induced potentials in cortical axons spread to other synaptic neurons, generating a propagation of neuronal activation in associated cortical and subcortical areas [17,18]. Using TMS, we assessed the cortical excitability of patients with EM and CM. In addition, only for the second group, we studied the changes in the neurophysiological data after treatment with BoNT/A.

Our results show that, in the group of patients affected by CM without aura, the values of motor threshold are significantly lower than those recorded in patients affected by EM without aura. This suggests greater cortical excitability in chronic migraine. Comparing the RMT, SICI, and ICF data of our patients with those of healthy subjects published in the literature [19], cortical hyperexcitability can be appreciated both in CM and EM patients.

Some previous studies, comparing patients with migraine (these studies did not distinguish between EM and CM or migraine with and without aura) and healthy controls, observed similar differences in motor and phosphene thresholds, which are suggestive of a condition of hyperexcitability of the motor and visual cortices, respectively [18,20,21,22]. However, to our knowledge, a specific comparison of TMS data between EM and CM is still lacking in the literature.

Analysing the pp-TMS data, we found a physiological inhibition (although reduced compared with healthy subjects) of the conditioned response for ISIs of 1–4 ms, whereas we observed a facilitated response for ISIs of 6–15 ms. These results are in line with previous studies [20,21,22]. In the group of patients affected by CM, before starting treatment with BoNT/A, the data showed a lack of physiological intracortical inhibition for short ISIs, whereas the intracortical facilitation was maintained and exaggerated for the 6, 10, and 15 ms intervals. Since the genesis of CM can be viewed as an incorrect response to environmental stimuli leading to the sensitisation of cortical areas, our results could be interpreted as a sign of the maintenance of a sensitisation of the trigeminovascular system—a phenomenon which is at the base of the painful phase of migraine. In a situation of reiterated stimulation of the trigeminal nerve, the pathophysiological events that occur during a single migraine attack could cause the perpetuation of a facilitation of the nociceptive transmission to the caudal nucleus of the trigeminal nerve [23,24].

In chronic migraine, as in other conditions characterised by altered processing of nociceptive stimuli, pain signals from the periphery are transmitted to the CNS (trigeminal nucleus caudalis, etc.), where they can induce central sensitisation [14,25]. BoNT/A is an effective prophylactic treatment for CM [26]. Based on previous evidence, it was believed that BoNT/A could function only in peripheral nerve terminals, inhibiting the release of inflammatory neurotransmitters and mediators. Now it is known that BoNT/A can reach the central nervous system from the peripheral nerves by axonal transport so that it can inhibit the release of pain mediators in the spinal cord, brainstem, and brain. Thus, BoNT/A alters CNS nociceptive transmission and reduces central sensitisation [14,24,25]. 

After performing prophylactic therapy with BoNT/A, patients with CM presented a significant improvement in clinical status, confirmed by the decrease in MIDAS score. Based on sp-TMS data, we conclude that BoNT/A does not modify the corticospinal pathway excitability. Specifically, BoNT/A does not change RMT, CCT, or CSP. RMT reflects the excitability of fast-conducting pyramidal neurons; it is increased by sodium channel blocker administration but is not modified by drugs that alter GABAergic or glutamatergic conduction [27]. CMCT is a neurophysiological measure that reflects conduction between the primary motor cortex and spinal cord. With electrical stimulation, it includes the time for impulse propagation via the fast-conducting neurons in the corticospinal tract and excitation of the spinal motoneurons sufficient to exceed their firing threshold. Therefore, it reflects the activation of voltage-gated sodium channels of pyramidal neurons. The CSP principally reflects the activation of GABA_B_ receptors, so it increases progressively after the administration of GABAergic drugs such as baclofen [28]. 

Therefore, we hypothesise that the mechanism of action of BoNT/A in chronic migraine is not mediated by voltage-gated sodium channel binding or by interaction with GABA_B_ receptors.

From noteworthy pp-TMS studies, we found that botulinum toxin treatment was followed by a decrease of the paradoxical facilitation observed for ISIs of 1–6 ms. Although this change was not statistically significant (marginally for the 2 ms ISI only), the reliability of the *t*-test inference should be considered cautiously in this case because of the higher inter-subject variability, the different variance of measurements in the two groups, and the low number of patients. 

SICI likely represents post-synaptic inhibition mediated by GABA_A_ receptors because drugs that enhance GABA_A_ergic neurotransmission increase SICI [27,29]. For ICF, excitatory glutamatergic circuits in M1 may be involved, and it has been suggested that this form of facilitation might result from the recruitment of additional cortical circuits separate from those more easily activated by single-pulse stimulation [28]. Thus, we can speculate that BoNT/A could exert its effect through interaction with intracortical GABA_A_ and glutamatergic receptors.

Our results demonstrate that patients affected by CM present a change in cortical plasticity, which is not completely reversible after appropriate prophylactic treatment with BoNT/A, at least in the time frame of our study. Our results seem to indicate that treatment with BoNT/A may be insufficient, at least after a treatment period limited to one year, to reverse the neurophysiological changes associated with chronic migraine. In other words, our study raises doubts on the possibility of extinguishing the “fire” of migraine once it becomes chronic.

In clinical practice, prophylaxis of CM is recommended for a one-year term. This treatment duration was followed in our protocol. It is possible that the minor changes in cortical plasticity observed in our study may be extended and consolidated with a longer duration of treatment.

## 4. Conclusions

It is well known that migraine is a disorder characterised by altered sensory processing. Our study demonstrates that patients suffering from CM present an altered response to pp-TMS, confirming the hypothesis of an alteration of cortical plasticity due to chronic pain. The core of synaptic plasticity is the reshaping of the excitatory–inhibitory balance through modifications of synaptic weights occurring in both excitatory and inhibitory synapses. Our patients presented a facilitated response even for ISIs in which an inhibition occurred, showing an imbalance between excitatory and inhibitory networks.

The second important element is that this type of pathological response was still observable, even if reduced, after appropriate prophylactic treatment, despite the fact that BoNT/A led to a marked clinical improvement and the disappearance of chronic pain.

We think that the mechanisms of cerebral maladaptation to environmental sensory stimuli, which are triggered by chronic pain, lead to plasticity changes that persist even after the pain is interrupted. The results of our study support the hypothesis that the changes due to chronic pain act on both synaptic and anatomic plasticity. A major limitation of our study is the small sample size; moreover, we did not assess potential confounders, such as the presence of comorbid psychiatric disorders. Indeed, it has been demonstrated that anxiety and affective disorders, as well as PTSD, are more frequent in migraineurs than in the general population, with a higher prevalence in patients with CM than in patients with EM. 

Several mechanisms have been proposed to explain the comorbidity of migraine and psychiatric disorders, including unidirectional and bidirectional causal models, shared environmental or genetic risk factors, and latent brain state models [30].

Affective and other mental disorders can modify cortical excitability parameters. As shown in a recent systematic review by Bunse et al., patients displayed a general pattern of cortical disinhibition, rather than disease-specific changes, across several different psychiatric diseases. Only schizophrenia, obsessive-compulsive disorder, and Tourette syndrome showed reproducible reduced SICI values, whereas affective and other mental illnesses showed extremely variable results, making definitive conclusions unreliable. This may be due to the heterogeneous phenotype of affective disorders [31].

Regarding post-traumatic stress disorder (PTSD), one study by Rossi et al. showed a reduced SICI in the left but not in the right hemisphere, while ICF was increased in the right but not in the left hemisphere [32]. However, it should be emphasised that none of our patients underwent specific pharmacological treatment, which could have modified the potential psychiatric comorbidities during the observation period. Therefore, any confounding factors regarding the effects of psychiatric comorbidities on cortical excitability parameters would have been the same between the pre-BoNT/A and post-BoNT/A conditions. However, it could be hypothesised that BoNT/A treatment, due to its beneficial effect on the severity of pain, may have also reduced potential comorbid conditions, such as anxiety or affective disorders, with subsequent modifications of cortical excitability. Indeed, a recent clinical trial showed that BoNT/A treatment for CM was associated with clinically significant reduction in depression and anxiety as well as an improvement in sleep quality and fatigue [33].

Further studies are required to demonstrate if these changes are maintained and increased by longer treatment with BoNT/A.

## 5. Materials and Methods

### 5.1. Study Population

Patients affected by CM and EM, defined according to the International Classification of Headache Disorders (ICHD-3) [5], were recruited from the Headache Centre of our Clinical Neurology Unit from September 2014 to September 2017. Exclusion criteria were as follows: patients < 18 years old, patients not meeting the diagnostic criteria for CM or EM, patients presenting contraindications for TMS (in particular, psychiatric disorders, epilepsy, and cardiac implantable devices), patients treated with antiepileptic drugs and/or benzodiazepines, patients with EM under prophylaxis therapy with any drug at the time of evaluation or who had been treated in the previous 3 months, patients refusing to consent to the study.

For all patients, we collected clinical data about the characteristics of migraine, pharmacological history, and MIDAS (Migraine Disability Assessment Score) questionnaire [34].

### 5.2. BoNT/A Treatment Protocol

Patients affected by CM were treated with BoNT/A (onabotulinumtoxin A, BOTOX^®^, Allergan^®^) every 12 weeks for 4 cycles in accordance with the Phase III Research Evaluating Migraine Prophylaxis Therapy (PREEMPT) injection paradigm [35,36]. All patients received a minimum intramuscular dose of 155 IU of BoNT/A administered to 31 injection sites across 7 head and neck muscles using a fixed-site, fixed-dose injection paradigm (each injection was 5 IU in 0.1 mL saline). In addition, using a follow-the-pain approach, intramuscular administration of up to 40 IU BoNT/A was allowed in additional injection sites across the head and neck muscles. Thus, the minimum dose in an individual patient was 155 IU, and the maximum dose was 195 IU. 

### 5.3. TMS Protocol

TMS studies were performed using MagPro^®^ magnetic stimulator (MagVenture Inc., Alpharetta, GA, USA) connected to an electromyographic device (Dantec™ Keypoint^®^, Natus^®^, Middleton, WI, USA). Cortical excitability was evaluated by means of well-established sp-TMS and pp-TMS protocols in line with the International Federation of Clinical Neurophysiology guidelines [28,37]. Patients affected by CM underwent TMS protocol at enrolment (baseline) and one year after repeated injections of BoNT/A; patients affected by EM were administered the same protocol only at enrolment.

All TMS studies were performed with a figure-of-eight coil, which was placed on the scalp region corresponding to the motor cortex contralateral to the examined upper limb (hand area). The coil was oriented 45° towards the contralateral forehead in order to guarantee current flow approximately perpendicular to the central sulcus [28].

The motor evoked potentials (MEPs), obtained from single magnetic stimuli, were recorded from surface electrodes located on the first dorsal interosseous muscle of each side. LFF was set at 20 Hz; HFF was set at 10 kHz.

From sp-TMS sessions, the following parameters were collected:(a)Rest motor threshold (RMT), defined as the minimum stimulus intensity which, in a sequence of 20 stimuli, can evoke MEPs with peak-to-peak amplitudes > 50 μV in at least 50% of trials (10 out of 20 trials) [38].(b)Cortical silent period (CSP), defined as the period of electrical silence in the surface EMG activity that occurs immediately after the MEP when a focal suprathreshold TMS is delivered to the motor cortex during a tonic muscle contraction [28].(c)Central motor conduction time (CMCT), calculated by subtracting the peripheral conduction time from the latency of MEPs evoked by transcranial cortical stimulation. Peripheral conduction time was estimated by taking half of the result of adding the F wave latency and the M wave latency to nerve stimulation (cathode proximal) and subtracting 1 ms, i.e., (F + M − 1)/2 [28].

For CMCT calculation, we used the maximal MEP peak-to-peak amplitude, obtained by delivering magnetic stimuli of 140% RMT intensities. The efficacy of TMS in exciting the corticomotor output was increased by asking the patient to pre-activate the target muscle at 10–20% of maximum strength. For each muscle, 5–6 consecutive MEPs were recorded during tonic contraction [37].

Moreover, from pp-TMS sessions, the following parameters were collected [39]:(a)Short interval intracortical inhibition (SICI), elicited by delivering a subthreshold (80% RMT) conditioning stimulus (CS) followed by a suprathreshold (120% RMT) test stimulus (TS) at interstimulus intervals (ISIs) of 1, 2, 3, and 6 ms.(b)Intracortical facilitation (ICF), evoked with a similar protocol as SICI but at longer ISIs of 10 and 15 ms.

For both SICI and ICF, the MEP peak-to-peak amplitudes were compared to those produced by the TS alone as a reference (control) condition. The size of the conditioned responses was expressed as a percentage of the size of the control response. The mean response between the right and left hemispheres was reported. As these studies should be done with the target muscle at rest, background EMG activity was monitored and recorded to determine the state of muscle relaxation.

### 5.4. Statistical Analysis

Descriptive analysis of the main features of the study population was performed by using mean ± SD or percentages for categorical variables.

For the statistical analysis, a *t*-test for paired data was used to compare neurophysiological variables (RMT, CSP, CMCT, SICI, and ICF) in patients affected by CM between baseline (enrolment) and one year after BoNT/A treatment for both hemispheres. A *t*-test for unpaired data was used to compare the same variables between CM patients and patients affected by EM at baseline.

All analyses were conducted using Stata/SE (version 14.0 Stata Corp., College Station, TX, USA) for Mac. All 2-tailed statistical significance levels were set at *p* < 0.05.

The study protocol was approved by our local Ethics Committee (Name: Department of Medical and Experimental Sciences, University of Udine, Internal Review Board; Approval number: 19/2014, approved on 11 July 2014).

## Figures and Tables

**Figure 1 toxins-13-00392-f001:**
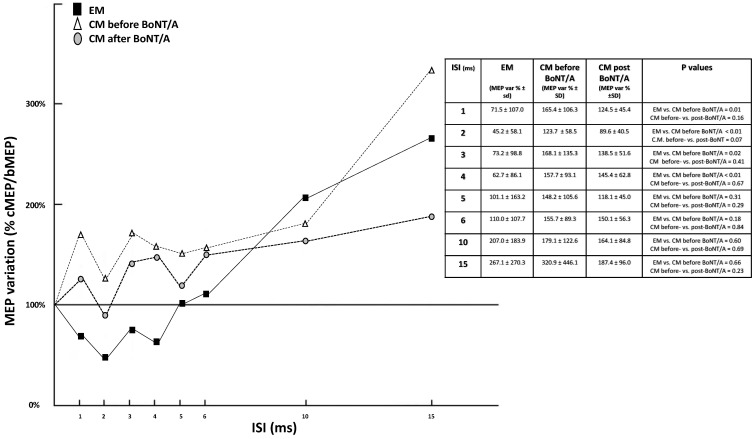
Cortical excitability parameters obtained from pp-TMS in EM patients, CM patients before and after BoNT/A treatment. Legend: cMEP, conditioned motor evoked potential; bMEP, basal motor evoked potential; pp-TMS, paired-pulse transcranial magnetic stimulation; pp, paired-pulse; ISI, interstimulus interval; EM, episodic migraine; CM, chronic migraine. Notes: MEP variation is reported as the mean percentage of variation cMEP/bMEP between right and left hemispheres. The CM group lacks physiological SICI and, on the contrary, has facilitation for lower ISIs. After BoNT/A treatment, SICI appears partially reduced, though not significantly. The chart does not display standard deviation (SD) in order to make each single group trend more readable; the table beside it shows the percentage of MEP variation. A t-statistic *p*-value is reported for EM vs CM before BoNT/A treatment as well as for CM before BoNT/A treatment and CM after BoNT/A treatment.

**Table 1 toxins-13-00392-t001:** Comparison of cortical excitability parameters obtained from sp-TMS between EM and CM before BoNT/A treatment (A) and between CM before and after BoNT/A treatment (B).

	**Left hemisphere**			**Right hemisphere**		
**A**	**RMT (%MSO)** **mean ± SD**	**CSP (ms)** **mean ± SD**	**CMCT (ms)** **mean ± SD**	**RMT (%MSO)** **mean ± SD**	**CSP (ms)** **mean ± SD**	**CMCT (ms)** **mean ± SD**
**Episodic Migraine**	53.54 ± 6.58	91.11 ± 58.70	6.39 ± 1.25	52.00 ± 4.28	91.9 ± 36.84	7.86 ± 1.62
**Chronic Migraine** **before BoNT/A**	45.54 ± 8.10	68.04 ± 33.4	7.17 ± 1.55	46.00 ± 7.80	80.80 ± 44.78	7.54 ± 1.24
***p*-value**	0.02 *	0.29	0.89	0.03 *	0.61	0.21
	**Left hemisphere**			**Right hemisphere**		
**B**	**RMT (%MSO)** **mean ± SD**	**CSP (ms)** **mean ± SD**	**CMT (ms)** **mean ± SD**	**RMT (%MSO)** **mean ± SD**	**CSP (ms)** **mean ± SD**	**CMT (ms)** **mean ± SD**
**Chronic Migraine** **before BoNT/A**	45.54 ± 8.10	68.04 ± 33.40	7.17 ± 1.55	45.90 ± 7.84	80.81 ± 44.78	7.54 ± 1.24
**Chronic Migraine** **after BoNT/A**	44.57 ± 2.99	65.60 ± 17.57	7.72 ± 1.32	42.14 ± 3.93	58.64 ± 29.03	8.22 ± 0.99
***p*-value**	0.77	0.87	0.44	0.26	0.27	0.25

Legend: RMT, rest motor threshold; MSO, maximal stimulator output; CSP, cortical silent period; CMCT, central motor conduction time. * indicates *p*-value < 0.05.

## Data Availability

Detailed data is available upon request.

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
