# Peer review of "Clinical and Neurophysiological Effects of Botulinum Neurotoxin Type A in Chronic Migraine"

_toxins, 2021, doi:10.3390/toxins13060392_

Round 1

Reviewer 1 Report

This is a well written and interesting staudy about Migraine and TMS.

I have 3 minor points:

  1. Please indicate that the study was approved by the ethic committee.
  2. From my point of view and from patients I see every day, TMS is not "painless" (line 122), please correct it.
  3. this is a very small sample size, therefore, the limitations of the result should be mentioned in the discussion section.

Author Response

I have 3 minor points:

  1. Please indicate that the study was approved by the ethic committee.

The study was approved by our local IRB. We now have clearly reported IRB name and approval number in the back-matter of the manuscript.

  1. From my point of view and from patients I see every day, TMS is not "painless" (line 122), please correct it.

Diagnostic TMS is a non-invasive procedure, which is usually very well tolerated. In any case, if any annoyance may be elicited, this cannot be certainly described as “pain”.

  1. this is a very small sample size, therefore, the limitations of the result should be mentioned in the discussion section.

This remark is certainly correct. We emphasized the problem in discussing the limitations.

Reviewer 2 Report

This study investigated longitudinal changes in cortical plasticity and migraine-related disability in migraine patients receiving botox treatment.  

  1. No mention of sample size and study power estimation; please include sample size estimation. Or examine your effect size and study power post-hoc. 
  2. Please use longitudinal analytic approach i.e. linear mixed model to account for repeated measures adjustment: baseline to end of 4 botox treatment for CM
  3. How about assessment of common comorbidities such as anxiety, depression, PTSD etc that have known effects on migraine-related disability and cortical excitability changes. Show assessment of these confounders either by adjusting to comorbidities or by conducting a sensitivity analysis.
  4. Manuscript needs professional scientific English editing. There are multiple English errors. 

Author Response

This study investigated longitudinal changes in cortical plasticity and migraine-related disability in migraine patients receiving botox treatment.  

  1. No mention of sample size and study power estimation; please include sample size estimation. Or examine your effect size and study power post-hoc. 

During the study planning we did not perform an a-priori sample estimation, given the lack of specific data in CM. A post-hoc power analysis reveals low-to-moderate power (20 to 40%) for the ppTMS variation in CM - between before-treatment and post-treatment measurements - and a moderate to high power for spTMS  (71%) parameters between EM and CM. Regarding this last aspect (sp-TMS in EM vs CM) , we think that our sample size was sufficient. In pp-TMS measurements (pre- vs post-treatment in CM), we think that the effect size of our measurements in CM is clinically relevant and that further studies with larger sample size could reach significant results. 

Should the Editor consider worthwhile to add the above statements regarding post-hoc power analysis, we are ready to do that.   

  1. Please use longitudinal analytic approach i.e. linear mixed model to account for repeated measures adjustment: baseline to end of 4 botox treatment for CM

We agree with the reviewer that it would have been an interesting point to explore. Unfortunately, we did not collect repeated measurements during the year of BoNT treatment and we cannot provide this kind of analysis.

The protocol was very challenging and time consuming and we limited ourselves to observe TMS changes at baseline and at the end of the usual treatment period (four cycles of BoNT/A, as verified in the PREEMPT study). However, after the results of this first study, we will definitely take the reviewer suggestion for planning a future protocol on TMS and migraine.

  1. How about assessment of common comorbidities such as anxiety, depression, PTSD etc that have known effects on migraine-related disability and cortical excitability changes. Show assessment of these confounders either by adjusting to comorbidities or by conducting a sensitivity analysis.

Thank you for your observations. You are correct: affective and other psychiatric disorders do affect cortical excitability parameters. Indeed, as shown in a recent systematic review by Bunse et al, across several different psychiatric diseases, patients displayed a general pattern of cortical disinhibition rather than disease-specificic changes. Only schizophrenia, obsessive-compulsive disorder and Tourette syndrome showed repeatable reduced SICI values while in affective and other mental illnesses very variable results were found, making definitive conclusions almost unreliable. This can be due to the heterogeneous phenotype of affective disorders. [1]

Regarding PTSD, one study by Rossi et al showed a reduced SICI in the left but not in the right hemisphere while ICF was increased in the right but not in the left hemisphere.[2]

It has been demonstrated that anxiety and affective disorders as well as PTSD are more prevalent in migraineurs than in general population with higher prevalence among patients with CM compared to EM.[3]

Several mechanisms have been proposed to explain the comorbidity of migraine and psychiatric disorders, including unidirectional and bidirectional causal models, shared environmental or genetic risk factors, and latent brain state models. [3]

Unfortunately, we did not assess these issues in our sample, so we are not able to perform any further analysis in order to address these potential confounders. However, it should be emphasized that none of our patients underwent specific pharmacological treatment which could have modified the potential psychiatric comorbidities during the observation period. So, any confounding factors regarding the effects of psychiatric comorbidities on cortical excitability parameters would have been the same between the pre-BoNT and post-BoNT conditions.

However, it could be hypothesized that BoNT treatment, due to its beneficial effect on the severity of pain, may also have reduced potential comorbid conditions like anxiety or affective disorders with subsequent modifications of cortical excitability. Indeed, a recent clinical trial showed that BoNT/A treatment for CM was associated with clinically significant reduction in depression and anxiety, and with an improvement of sleep quality and fatigue. [4]Further studies on larger samples will be necessary to clarify these topics.On the other side, we profited of the reviewer’s comment to specify that the protocol used to study cortical excitability has nothing to do with the therapeutic use of TMS for mental disorders.

We have added these considerations in the revised manuscript.

  1. Bunse T, Wobrock T, Strube W, Padberg F, Palm U, Falkai P, et al. Motor cortical excitability assessed by transcranial magnetic stimulation in psychiatric disorders. A systematic review. Brain stimulation, 2014; 7: 158-169
  2. 2. Rossi S, De Capua A, Tavanti M, Calossi S, Polizzotto NR, Mantovani A, et al. Dysfunctions of cortical excitability in drug-naive posttraumatic stress disorder patients. Biol Psychiatry, 2009; 66: 54-61
  3. Buse DC, Silberstein SD, Manack AN, Papapetropoulos S, Lipton RB. Psychiatric comorbidities of chronic and episodic migraine. J Neurol, 2013; 260: 1960-1969
  4. Blumenfeld AM, Tepper SJ, Robbins LD, Manack Adams A, Buse DC, Orejudos A, et al. Effects of onabotulinumtoxinA treatment for chronic migraine on common comorbidities including depression and anxiety. J Neurol Neurosurg Psychiatry, 2019; 90: 353-260
  1. Manuscript needs professional scientific English editing. There are multiple English errors. 

We did a complete revision of the manuscript. English mistakes have been corrected.

Round 2

Reviewer 2 Report

Please include your post-hoc power analysis which you described in the study, as it is important. 

Author Response

The post-hoc power analysis has now been added in the results section of the manuscript.